cognition/psychology

animal vocalizations, pet-owners, emotion perception, cat miaows, dog whines, crying

**Author for correspondence:**
Christine E. Parsons
e-mail: christine.parsons@clin.au.dk

# Pawsitively sad: pet-owners are more sensitive to negative emotion in animal distress vocalizations

Christine E. Parsons[1], Richard T. LeBeau[2], Morten L. Kringelbach[3] and Katherine S. Young[2,4]

[1]Interacting Minds Center, Department of Clinical Medicine, Aarhus University, Aarhus, Denmark
[2]Department of Psychology, University of California, Los Angeles, CA, USA
[3]Department of Psychiatry, University of Oxford, Warneford Hospital, Oxford, UK
[4]Social, Genetic and Developmental Psychiatry Centre, Institute of Psychology, Psychiatry and Neuroscience, King's College London, London, UK

CEP, 0000-0003-2856-6308

Pets have numerous, effective methods to communicate with their human hosts. Perhaps most conspicuous of these are distress vocalizations: in cats, the 'miaow' and in dogs, the 'whine' or 'whimper'. We compared a sample of young adults who owned cats and or dogs ('pet-owners' $n = 264$) and who did not ($n = 297$) on their ratings of the valence of animal distress vocalizations, taken from a standardized database of sounds. We also examined these participants' self-reported symptoms of anxiety and depression, and their scores on a measure of interpersonal relationship functioning. Pet-owners rated the animal distress vocalizations as sadder than adults who did not own a pet. Cat-owners specifically gave the most negative ratings of cat miaows compared with other participants, but were no different in their ratings of other sounds. Dog sounds were rated more negatively overall, in fact as negatively as human baby cries. Pet-owning adults (cat only, dog only, both) were not significantly different from adults with no pets on symptoms of depression, anxiety or on self-reported interpersonal relationship functioning. We suggest that pet ownership is associated with greater sensitivity to negative emotion in cat and dog distress vocalizations.

## 1. Introduction

Approximately 57 per cent of US households reported owning a pet at the end of 2016, with cats and dogs being by far the most popular choices [1]. While more households reported having one or more

dogs (38%), considerable numbers also had cats (25%). Dog ownership is often viewed as healthy, and beneficial in motivating owners to get extra physical activity, 'The Lassie effect' [2], named after a heroic, fearless television collie. By contrast, cats and cat-owners are regularly ascribed negative personality traits.[1] Stereotypically, cat-owners are viewed as lonely, more emotional, and more depressed than dog-owners. A pet charity reported that about 50% of the Americans surveyed believed various long-standing stereotypes about cat-owners (especially the 'cat lady' image[2]).

Negative characterizations for those with an affinity for cats are not a recent phenomenon. One *New York Times* editorial from 1872, headlined 'Cats and Craziness', lays out a portrait of an infatuated cat-lover, differentiated from the more rationally behaved dog-lover. While these ideas persist, studies to support the idea of personality differences between cat- and dog-owners have been sparse. One online study of more than 4000 adults recruited from a range of countries, reported on the Big Five Personality traits of adults self-identifying as 'cat people' or 'dog people' (but not necessarily owning a cat or dog). 'Cat people' scored higher on measures of Neuroticism and Openness than 'dog people', but lower on Extraversion, Agreeableness and Conscientiousness [3].

Beyond the Big Five, another online study of 1000 primarily US adults found that 'dog people' were more socially dominant and competitive than 'cat people' (but there was no difference between the pet-owners on narcissism [4]). Effect sizes were small, but again, apparent even when asking about cat or dog affinity, rather than ownership. Since social dominance is associated with political conservatism [5], it is plausible that self-categorized 'dog people' are more likely to identify as conservative. A report from the American Veterinary Medical Association [6] suggested that this is indeed the case. They surveyed pet-owners from the 10 US states with households with the highest and lowest rates of dog and cat ownership. Nine of the top 10 dog-owning states voted Republican in the 2012 Presidential Election, and 9 of the bottom 10 dog-owning states voted Democrat. This was not the case for cat-owners: the top and bottom 10 cat-owning states were both split equally for Republican and Democrat candidates.

Two studies using social media data to analyse the behaviour of 'cat people' and 'dog people' also suggest some differences between the two types of pet-owners. Facebook published an analysis of data from 160 000 US users who posted images of either cats or dogs on their site. Those users who posted cat photos (i.e. the 'cat people') were more likely to be single than dog people, based on their Facebook relationship status. They also had 26 fewer Facebook friends than dog people, although they did receive more invitations to events.[3]

A second study examined the Facebook updates of adults who posted statuses about animal ownership ('my cat' or 'my dog') and who had previously filled in the International Personality Item Pool proxy for the NEO Personality Inventory Revised (NEO-PI-R). Facebook users mentioning 'my cat' or 'my dog' were slightly lower in conscientiousness than the general population. Users mentioning their cats (specifically using the term 'my cat') were more neurotic, open to experience, and introverted compared to users who did not. Users mentioning their dogs, however, did not emerge as having any other specific personality traits [7].

## 1.1. Perception of emotion in cats and dogs

While there is evidence for some personality differences between people identifying as 'dog' and 'cat' people, other work suggests that there are also differences between the way people who own these pets perceive and interact with them. Surveys of pet-owners in both Japan and Holland have found that dog-owners report noticing both joy and sadness more often in their animals than cat-owners [8,9]. Martens *et al.* discussed this as an illustration of the difference in dogs' and cats' communicative tendencies, referring to the social nature of dogs as pack animals and cats as semi-solitary animals. It is possible that dogs and cats are inherently different in how frequently or intensely they signal to humans. Indeed, cats are often valued for their independent personalities, whereas dogs are valued for their unconditional love [10].

## 1.2. Cat miaows and dog whines

Pets have numerous, effective methods to communicate with their human hosts. Perhaps most conspicuous of these are distress calls: in cats, the 'miaow' and in dogs, the 'whine' or 'whimper'. Dogs are the oldest

---

[1]Number of pets in the United States in 2017/2018, by species (in millions) [Internet]. The Statistics Portal 2018

[2]Daily Pilot. My pet world: join the campaign to debunk tired stereotypes about cats. https://www.latimes.com/socal/daily-pilot/entertainment/tn-hbi-et-0618-my-pet-world-20150618-story.html; 2015.

[3]Cat People, Dog People [Internet]. 2016.

domesticated companion animal [11], and are remarkably attuned to human social signals [12]. In turn, human listeners are skilled at interpreting cues from dogs. For instance, adults can guess, with reasonable accuracy, the context in which a dog bark occurs, scoring above chance in one study (40% accuracy rate in a six category choice task [13]). Dog ownership, or even experience with the specific breed of dog examined, did not affect the accuracy of guesses. Furthermore, domestication has impacted the vocal repertoire of dogs [14]. For example, dogs now use certain signals in contexts where they originally did not occur, such as barking to seek contact when they are isolated [15].

A similar domestication effect is also evident for cats: miaowing is not observed in adult wild cats and is a signal directly aimed at humans [16]. However, counter to findings for dog barks, human listeners seem to be relatively poor at categorizing cat miaows (scoring just above chance when categorizing unfamiliar cats [17,18]). In both studies, experience with cats was associated with improved accuracy. In the Ellis *et al.* study [18], the 10 cat-owners tested also performed better when listening to their own cat, relative to an unfamiliar cat. For cats at least, the small body of available evidence suggests that owning a cat can improve the interpretation of cat distress signals.

For dogs, most of the work has focused on barks [13,19,20], and ownership seems to impact the accuracy of interpretation only minimally. Beyond barks, growls are one signal where dog ownership has been associated with increased accuracy of categorization (although having been bitten by a dog did not [21]). The authors of this study argued that differentiation of growls might require more experience than barks, which are a loud, long-range and common sound.

Despite the prevalence of pet owning, and the impact of domestication on cat and dog vocal patterns, relatively few studies have examined how humans interpret the emotional content of pet vocalizations. Furthermore, most studies to date on the characteristics associated with pet ownership have also focused on classic personality traits (e.g. the Big Five). In this work, we had two major aims: (i) We compared pet-owners and non-pet-owners on measures of depression and anxiety symptoms, along with a measure of adult attachment. We hypothesized that cat-owners would show greater anxiety and depression symptoms compared to dog-owners and adults with no pets. As a secondary analysis, we examined the associations between symptoms of depression, anxiety and adult attachment and perception of animal vocal sounds. (ii) We compared adults with and without pets on their perception of the emotional content of both dog and cat vocal signals. We hypothesized that pet-owners would show greater sensitivity to emotion in animal vocal sounds than adults without pets. We tested a large sample of young adults using sound stimuli from a standardized database of vocal sounds that included both cat and dog vocalizations, as well as human distress vocalizations (the OxVoc Sounds Database [22]). This also allowed us to compare the same participants' ratings of animal sounds with other sounds that elicit caregiving (infant cry vocalizations) and sounds signalling distress (adult cry vocalizations, main findings reported in [23]).

## 2. Method

### 2.1. Participants

Participants were recruited using online advertising from the University of California, Los Angeles (UCLA) Psychology Department Subject Pool of undergraduate students. Participants received course credit for taking part in the study.

In total, 561 (111 males) participants took part in the present study on animal vocalization perception. This subsample comprised the first wave of participants who took part in a larger study ($N = 945$) to validate an openly available vocal sounds database (the full sample provided adequate power to examine test–retest reliability and categorization validity; Oxford Vocal 'OxVoc' Sounds Database [23]). Participants were young adults (age: $M = 20.22$ years, s.d. $= 2.1$, range $= 18–37$) and most did not have children ($n = 5$ parents). Participants self-reported as Asian ($n = 253$), Caucasian ($n = 130$), Hispanic/Latino ($n = 84$), and a range of other racial/ethnic backgrounds ($n = 95$). Nearly half of the recruited sample said they owned either a cat or dog (47%), referred to throughout as 'pet-owners'. The majority owned a dog only ($n = 184$), and a smaller number owned a cat only ($n = 31$) or both a cat and a dog ($n = 49$).

### 2.2. Self-report measures

Symptoms of depression were assessed using the Edinburgh Depression Scale (the EDS, based on the Edinburgh Postnatal Depression Scale; a measure with high specificity and sensitivity [24]). We used

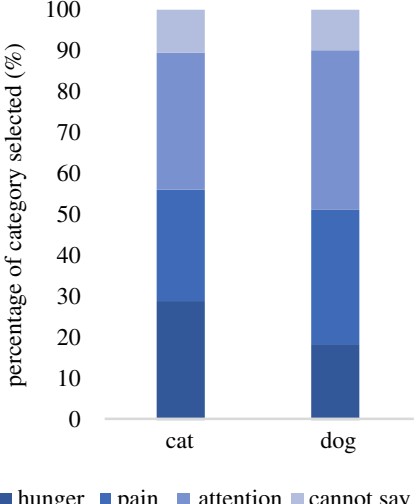

**Figure 1.** Results of the categorization task across the cat and dog vocalizations. The perceived reasons for the sound had similar distributions for the two animal categories, and the most commonly selected reason was 'seeking general attention'.

9 of the 10 items from the EDS [25], removing the suicidality item. Versions of the EDS excluding the suicidality item (EDS-5 [26]) have previously shown that this item does not substantially contribute to explaining the variance of the full scale.

The Generalized Anxiety Disorder Questionnaire-IV (GAD-Q-IV) was used for assessing symptoms of anxiety. The GAD-Q-IV is a nine-item self-report measure with high sensitivity and specificity for detecting symptoms of generalized anxiety disorder (GAD), based on the Diagnostic and Statistical Manual of Mental Disorders, Fourth Edition (DSM–IV) criteria [27].

An 18-item version of the Experiences in Close Relationships (ECR) measure [28] was used to evaluate adult attachment levels. The ECR provides a continuous measure of attachment, rather than categorizing adults into an attachment style [29]. Respondents used a seven-point, partly anchored, Likert-type scale ranging from 1 (disagree strongly) to 7 (agree strongly) to respond to the items. Higher scores on this scale indicate more insecure attachments, whereas lower scores indicate more secure attachments.

## 2.3. Stimuli

All sound stimuli were from a standardized database of vocal sounds (the OxVoc Sounds Database [22]). This database contains affective vocal sound stimuli from human adults and infants and domestic animals. Domestic animal vocalizations ($n = 30$, 15 cat and 15 dog sounds) were obtained from a freely available online website, as were adult distress vocalizations ($n = 19$). Infant distress vocalizations ($n = 21$) were obtained from video recordings of infants interacting with carer in their own homes. All stimuli were free from background noise and were selected to ensure a duration of 1–2 s (see table 2 for physical properties; for database details, see [22]). Stimuli were matched for total root-mean-square amplitude for each clip to −25 dBFS (decibels full scale).

We did not have information on the reason for each animal's vocalization (if they were in 'hunger', 'pain' etc.) or the age of the animal. We therefore carried out a *post hoc* experiment using an online survey platform (Prolific) to examine the perceived reason for the animal's distress and the animal's age for the cat and dog vocalizations. Participants ($N = 31$) were asked to categorize the reason for the vocalization (four-alternative forced choice: hunger, pain, seeking general attention or cannot say) and the age of the animal (two-alternative forced choice: puppy versus dog on dog trials, kitten versus cat on cat trials, see electronic supplementary material for a full task description). The order of these two questions, and the stimulus presentation order, were randomized across participants. A $\chi^2$ test indicated that there were no significant differences in the perceived reasons for the cat and dog vocalizations (figure 1; $\chi^2$ test = 2.95; $p = 0.40$). The most common reason selected was 'seeking general attention' for both dogs and cats (see electronic supplementary material, figure S1 for data for each stimulus). There were also no significant differences in the number of sounds categorized as kitten or puppy ($\chi^2$ test= 0.10; $p = 0.76$).

**Table 1.** Depression (EDS), anxiety (GAD-Q) symptoms and experiences in close relationships (ECR) scores in adults with no pets, dogs only, cats only or both.

| pet status | EDS scores | | GAD-Q scores | | ECR scores | |
|---|---|---|---|---|---|---|
| | M | s.d. | M | s.d. | M | s.d. |
| no pet (n = 277) | 8.65 | 4.14 | 5.17 | 2.95 | 3.97 | 0.52 |
| dog only (n = 183) | 8.57 | 4.41 | 5.11 | 3.26 | 3.95 | 0.59 |
| cat only (n = 31) | 8.20 | 4.95 | 4.73 | 3.44 | 3.88 | 0.45 |
| cat and dog (n = 48) | 7.98 | 4.43 | 5.39 | 3.31 | 3.93 | 0.46 |

## 2.4. Procedures

Stimuli were presented using Presentation® software (Neurobehavioral Systems, Inc., Berkeley, CA, USA, www.neurobs.com) on a desktop PC. Sounds were presented through headphones at a volume that was comfortable for each participant. After listening to each sound, participants used a vertical visual analogue scale (VAS) to rate the valence. Instructions to participants were 'Please indicate how happy or sad you think each sound is'. Responses were encoded on a continuous numerical scale, with a sensitivity of two decimal places from + 4.00 'very happy' to −4.00 'very sad'). The order of stimulus presentation was randomized for each participant. Participants listened to all sounds from the OxVoc database, but we focus our analysis here on responses to the animal and human distress sounds (adult cry $n = 19$, infant cry $n = 21$, cat miaow $n = 15$, dog whine $n = 15$). Following an introductory phase, during which participants practised the rating procedure, the task was fully automated. Participants had a maximum of five seconds to provide a response to each stimulus, after which they were automatically moved on to the next trial. Participants used the 'UP' and 'DOWN' arrows on the keyboard to make VAS ratings.

Participants were tested in person, with a maximum of four participants seated at different workstations in the same laboratory under the supervision of a research assistant. After the sound rating, participants completed the self-report questionnaires measures described above.

## 2.5. Statistical analysis

A one-way ANOVA with pet-owning status as a between-subjects factor was used to examine differences in the depression and anxiety symptom scores, as well as the ECR scores. We examined the associations between valence ratings and these psychological measures using Pearson's correlations. A general linear model (GLM) was performed with pet-owning status as a between-subjects factor and animal sound (cat, dog) as a repeated measures factor. Post hoc Tukey tests were also conducted. We compared how participants rated the valence of the animal sounds (cat, dog) with the human cry sounds (adult, infant) using a repeated measures GLM. There were no differences overall between men and women on their ratings of the cat vocalizations ($F_{1,555} = 0.01$, $p = 0.93$, $\eta^2 = 0.00$) or dog vocalizations ($F_{1,555} = 0.65$ $p = 0.42$, $\eta^2 = 0.001$), so we did not include gender as a factor in any of our analyses.

## 3. Results

### 3.1. Depression, anxiety and experiences in close relationships in adults with and without pets

Most participants ($n = 436$, 77.4%) scored below the cut-score on the EDS ($M = 8.56$, s.d. = 4.30; cut-off score of greater than 12 a modified score to account for item removal) and on the GAD-Q ($n = 489$, 88.2%; cut-off score of greater than 9.40, $M = 5.11$, s.d. = 3.10).

Table 1 presents the depression and anxiety symptom scores for adults with and without pets. There were no significant differences between participants who owned pets (collapsing across all pet-owners, as this resulted in two similar large groups) and those who did not on either their self-reported depression (EDS; pet-owners: $M = 8.42$, s.d. = 4.47; no pets: $M = 8.70$, s.d. = 4.15; $F_{1,557} = 0.58$, $p = 0.45$, $\eta^2 = 0.00$) or anxiety symptoms (GAD-Q; pet-owners: $M = 5.10$, s.d. = 3.30; no pets $M = 5.10$, s.d. = 2.90; $F_{1,559} = 0.00$, $p = 0.99$, $\eta^2 = 0.00$). There was no significant difference between participants who owned pets and those who did not on their ECR scores (pet-owners $M = 3.94$, s.d. = 0.55; no pets $M = 3.96$,

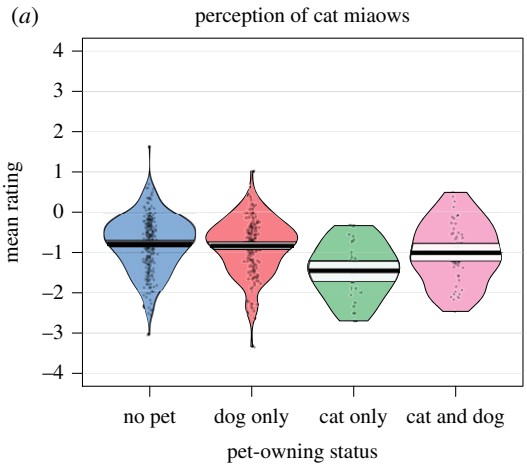
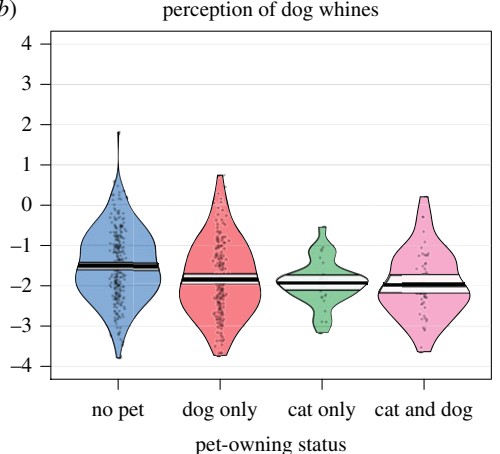

**Figure 2.** Pirate plot presenting valence ratings from the four participant groups for (*a*) cat and (*b*) dog vocalizations. (*a*) Cat-owners rated cat miaows significantly more negatively than all the other groups. (*b*) All pet-owners (dog only, cat only, cat and dog) rated the dog whines significantly more negatively than those who did not own pets. Raw data are represented by the black dots, the horizontal bar represents the mean, the coloured regions show smoothed densities and the rectangle shows the confidence intervals.

s.d. = 0.52; $F_{1,559} = 0.46$, $p = 0.50$). Table 1 presents the scores for these three questionnaires, for each of the different pet status groups. An additional comparison with the four participant groups resulted in the same pattern of results on these self-reported measures. There were no group differences in depression scores ($F_{3,535} = 0.40$, $p = 0.75$, $\eta^2 = 0.00$), anxiety scores ($F_{3,537} = 0.30$, $p = 0.82$, $\eta^2 = 0.00$), or ECR scores ($F_{3,537} = 0.26$, $p = 0.85$, $\eta^2 = 0.00$).

## 3.2. Depression, anxiety and ECR: associations with vocal ratings

There was no significant association between depressive symptoms and ratings of emotion in the cat (Pearson's $r = -0.02$, $p = 0.71$) or dog sounds (Pearson's $r = 0.00$, $p = 0.98$). There was also no significant association between anxiety symptoms and ratings of emotion in the cat (Pearson's $r = -0.03$, $p = 0.53$) or dog sounds (Pearson's $r = -0.04$, $p = 0.35$).

There was a small, positive correlation between ECR scores and valence ratings for the cat miaows ($N = 559$, $r = 0.10$, $p = 0.02$). People with less secure relationship attachments, as indicated by higher ECR scores, gave more positive ratings for the cat sounds. For the dog sounds, the direction of the association was the same, but not significant ($N = 559$, $r = 0.08$, $p = 0.06$).

## 3.3. Pet-owners rate animal distress calls as more negative than non-pet-owners

A GLM (animal sound, pet-owning status) showed a large main effect of the animal being rated, ($F_{1,557} = 573.81$, $p < 0.01$, $\eta^2 = .51$), a small significant main effect of pet-owning status ($F_{1,557} = 21.30$, $p < 0.01$, $\eta^2 = 0.04$), and a small significant interaction between the two ($F_{1,557} = 6.42$, $p = 0.01$, $\eta^2 = 0.01$). Pet-owners rated both cat miaows and dog whines as more negatively valenced compared to adults with no pet (figure 2). For the cat miaows, cat-owners provided the most negative ratings of the four participant groups. Their ratings were significantly more negative than dog-owners (Tukey *post hoc* test, $p = 0.03$), cat- and dog-owners ($p < 0.01$) and adults with no pet ($p < 0.01$). For the dog whines, all pet-owners provided more negative ratings than adults with no pets, but there were no significant differences between the pet-owning groups. The cat-owners were therefore distinctive in providing especially negative ratings for the cat miaows.

## 3.4. Dog whines are perceived as especially sad

Overall, the dog whines were rated as significantly sadder than the cat miaows (dogs: $M = -1.68$, s.d. = 0.90; cats: $M = -0.84$, s.d. = 0.71), and this was a large effect size ($\eta^2 = 0.51$). One acoustic feature that affects how we perceive sadness in human vocalizations is pitch. We, therefore, compared the estimated fundamental frequency (F0, Hz, related to pitch, as reported in [23]) of the different sounds (table 2). The mean F0 of the

**Table 2.** Key physical parameters of vocalizations for the animal sounds.

| | F0 (Hz) (M, s.d.) | burst duration (s) (M, s.d.) | no. of bursts (M, s.d.) | peak amplitude, dBFS (M, s.d.) |
|---|---|---|---|---|
| cat miaow | 406.98, 126.88 | 1.21, 0.35 | 1.27, 0.46 | −7.77, 1.74 |
| dog whine | 471.58, 54.93 | 0.82, 0.41 | 2.00, 0.85 | −7.43, 2.56 |

dog whines and the cat miaows were not significantly different from one another ($t_{28} = -1.80$, $p = 0.08$). There was also no significant correlation between mean ratings of valence for each sound (cat, dog) and its pitch ($N = 30$, Pearson's $r = -0.18$, $p = 0.35$).

Finally, we compared adults' ratings of the animal sounds with the human distress sounds. Again, there was a large significant effect of sound category ($F_{3,1671} = 486$, $p < 0.01$, $\eta^2 = 0.46$). Adult human cries received the saddest ratings ($M = -2.17$, s.d. $= 0.85$) followed by the human infant cries ($M = -1.69$, s.d. $= 0.80$) and dog whines ($M = -1.68$, s.d. $= 0.90$) and finally the cat miaows ($M = -0.85$, s.d. $= 0.70$). Overall, dog whines were rated as less sad than human cry sounds (collapsing across both infant and adult cries), but this effect was driven by ratings of the adult cries. We found no difference between the ratings of dog whines and human infant cries ($t_{557} = 0.31$, $p = 0.76$). An equivalence test, using the two one-sided tests (TOST) procedure [30], indicated that the observed effect size (dz $= 0.01$) was significantly within the equivalent bounds of $-0.4$ and $0.4$ scale points (or in Cohen's dz: $-0.29$ and $0.29$), $t_{557} = -6.7$, $p < 0.01$).

### 3.5. No differences in cat-owners' perception of distress in other sounds

Given their more extreme ratings of cat miaows, we tested whether cat-owners were more sensitive to vocal signals of distress more generally. We compared cat-owners with the other participant groups on their ratings of human crying (from adults and infant) and found no differences ($F_{3,536} = 0.03$, $p = 0.99$, $\eta^2 = 0.00$). From the OxVoc database, the greatest variability in participants' valence ratings was for the neutral adult human vocal sounds [23]. We also found no difference across the four participant groups in their responses to the valence of the neutral sounds ($F_{3,536} = 1.02$, $p = 0.38$, $\eta^2 = 0.01$).

## 4. Discussion

### 4.1. Depression, anxiety and adult attachment: small to null effects

Comparing adults who owned pets and those who did not, we found no evidence for any differences on our measures of depression, anxiety or adult attachment. These null findings are at odds with a body of studies suggesting differences in personality traits between cat-owners, dog-owners and adults who do not own pets (e.g. [3,4,7]).

We also found no correlation between ratings of the animal sounds and symptoms of anxiety or depression, but we did find a small, significant correlation between scores on the questionnaire measuring adult attachment (Experiences in Close Relationships) and ratings of the cat miaows. Adults with more secure relationship attachments gave more negative ratings (were more sensitive to distress) for the cat sounds. The direction of effects was similar for the dog sounds and relationship attachments, but was not significant. Previous studies have typically examined adult attachment security in relation to interpreting human infant signals. For example, one study found that adults with less secure attachment had greater difficulties identifying infant emotions, and were more likely to be amused or neutral in response to infant distress than adults with more secure attachment [31]. Adults' attachment security may also be associated with differences in the perception of other types of emotional stimuli, such as those from animals, albeit with small effect sizes.

### 4.2. Pet ownership and perception of animal vocalizations

We found several subtle differences between how adults with and without pets generally rated animal vocalizations. Adults with pets rated the valence of the animal distress vocalizations as sadder than adults without pets, suggesting that pet-owners are more sensitive to the emotional content of animal

vocal signals. We also found that cat-owners were especially attuned to distress in cat vocalizations. Adults who owned a cat (only) rated the cat miaows as sounding sadder (i.e. more distressed) compared with the other pet-owners and adults with no pet. Dog-owners, by comparison, gave similar ratings for the dog whines compared with all other pet-owners.

Comparing the two types of animal vocalization, dog whines were rated as sadder than cat miaows. In fact, ratings for the dog whines were as negative as those for infant cry sounds, as confirmed using a statistical equivalence test [30]. Very few of our participants were parents ($n = 5$), and far more of the total sample reported owning dogs ($n = 184$). These differences in experience (owning a pet but no experience of parenting) might account for why the included dog vocalizations were rated as negatively as crying human babies. Investigating the potential acoustic differences that might be associated with differences in ratings for cat and dog sounds, we found no clear pitch differences between these sounds that might account for the perceived valence difference. The dog sounds were on average slightly, but not significantly, higher pitched. Other acoustic properties that might impact the evaluation of distress, such as 'roughness' [32] or the number of silences within a vocalization [33], may be of interest in future studies.

One explanation for the variance in perceived negative valence between cat and dog vocalizations may be related to the differences in human care required by dogs and cats. Dogs are generally more dependent on their owners for care than cats, and therefore require an especially effective set of communicative signals. Furthermore, dog-owners have a greater tendency to anthropomorphize (ascribe human-like emotions) than cat-owners [34]. If this tendency extends to non-dog-owners, it might explain why the dog whines were a particularly plaintive sound for humans, as negative as a baby's cry for the tested participants.

For example, an experimental study of interspecies communication compared cats and dogs on their behaviour on a food access task [10]. Whereas dogs looked to the humans and back to the hidden food when they were unable to get access to it themselves, the cats persisted. They tried to get the food themselves and rarely looked to the human face. This work suggests that there are differences in between the ways in which cats and dogs communicate, or use gaze, with humans. We suggest that dog vocalizations are also potentially more emotionally negative for human listeners than cat miaows. Further analyses, on a broader range of dog and cat distress sounds, are required to test the hypothesis that dog whines are more likely to elicit emotional responses than cat miaows.

Dogs are frequently described as 'man's best friend', whereas cats are considered only semi-domesticated [35]. Our finding that dog whines sounded more negative than cat sounds to our participants may be considered within the context of our general treatment of cats and dogs. For instance, while the raw number of pet cats in the US is greater than dogs, cats are less likely to be adopted than dogs in pet shelters [36]. Indeed, more pet-shelter cats are euthanized that dogs, across countries (for review, see [37]). Our societal interest in dogs is so strong that one analysis argued that an effective method to draw attention to an important news stories is to feature a dog, even tangentially. Although they did not examine whether cats generate similar news coverage, this work showed that stories featuring dogs were more likely to receive wider news coverage than stories that did not [38]. The 'dog effect' was roughly equivalent to whether a story warrants front- or back-page national news coverage in the *New York Times*.

We found no evidence to support the 'cat lady' stereotype: cat-owners did not differ from others on self-reported symptoms of depression, anxiety or their experiences in close relationships. Our findings, therefore, do not fit with the notion of cat-owners as more depressed, anxious or alone. We found that cat-owners perceived the cat miaows more negatively than other participants. This finding is broadly in line with work showing that cat ownership improves sensitivity to cat miaows [17,18]. The effect was highly specific to the cat sounds: cat-owners did not rate other sounds more negatively, suggesting that they are not simply more sensitive to negative emotion than non-cat-owners. Like the other participants, cat-owners did rate dog sounds as more negative than the cat sounds.

However, it is unclear why those who owned a cat only were distinct in rating cat miaows more negatively compared with those who owned both cats and dogs. We speculate that for adults who owned both, the dog is likely to be the dominant pet, requiring more caregiving (e.g. walks, more frequent trips outside). We suggest that the 'dominant dog' effect might be what separates people who chose a cat alone from those who own both. More broadly, dogs are more omnipresent in our environments than cats. There are numerous analyses to show that we seek out dog-related content more frequently on the Internet (Google Searches, video-searches, Instagram focused accounts) [39]. Even where we are not seeking out animal-related material, dogs are seen in TV commercials more frequently than cats [40].

## 4.3. Strengths, limitations and future directions

A strength of this work is that we recruited a large in-person sample, as opposed to online, giving us greater certainty about participant behaviour during the rating task. We did not specifically recruit participants based on animal ownership, rather we used a large, available student sample. This meant that participants were not taking part simply because of an interest in pet-specific research. However, this also meant that participants were undergraduate students and predominantly female, limiting the generalizability of the findings. Furthermore, student populations may show differing political leanings compared with the general US population, and this may impact upon pet ownership [6]. For instance, a 2016 Higher Education Poll reported that 35.5% of UCLA freshmen identified as 'liberal' [41] compared with 25% of the general US population, as reported in a Gallup poll [42]. University students have also been shown to have elevated levels of depression [43] and psychological distress compared with the general population [44]. Again, these demographic and psychological factors potentially limit the generalizability of our findings. A further limitation was that the size of the 'cat only' sample was small, although nearly half of the full sample owned pets.

Many adults categorize themselves as either a 'dog person' or a 'cat person' even if they do not have pets, or even intend to have a pet [45]. We did not ask participants if they viewed themselves as a cat or dog person, which may be of interest in future work, along with information on the duration of pet ownership and the breed owned. Therefore, the present research does not address the degree to which ownership versus affinity is associated with greater sensitivity to vocalizations in the respective animals. We also asked participants to rate their perception of the vocalization, and not the emotion the vocalization elicited in them.

We used 30 animal stimuli in total, which were selected from an online database based on their duration and quality (no background noise). We did not know either the actual age of the animal who made the sound, or the cause of the sound, which would be of interest to address in future work. Additionally, we chose to look at cat miaows because they are the most common cat-to-human vocalization [46]. While miaows are a cat's primary vocal communicator of distress, they also occur in other contexts, such as when cats are soliciting affection from their owners [17]. Dog whines, in contrast, arguably indicate distress more exclusively. If we are to make strong conclusions about dog sounds being stronger communicators of sadness to human listeners than cat sounds, a larger set of stimuli would be required. Cat miaows could also be collected during more defined behavioural contexts, where distress alone is communicated. It would also be of interest to look at the duration of pet ownership, as a proxy marker for familiarity with cat or dog vocalizations. Our previous work showed that duration of human motherhood was associated with increased brain activity specifically to baby sounds, suggesting that caregiving experience can attune responses to baby sounds [47]. It may be that pet caregiving experience attunes responses to pet sounds in a similar way.

## 5. Conclusion

We demonstrate that cat- and dog-owners are more sensitive than non-pet-owners to negative emotion in cat and dog vocalizations. This may be because pet-owners have more experience interacting with these animals, or because they are initially more responsive to these animals and therefore seek them as pets. Cat-owners rated cat miaows as more negative relative to the other adults with and without pets. We found no differences between cat-owners and the other participants on any of the self-report measures of anxiety, depression or experiences in relationships. We suggest that our findings are, therefore, not consistent with a description of cat-owners as depressed, anxious or as having difficulty with human relationships.

Ethics. Ethical approval was granted by the UCLA institutional review board and written informed consent was obtained from all participants.
Data accessibility. Data available from the Dryad Digital Repository: https://doi.org/10.5061/dryad.ss27j10 [48].
Authors' contributions. C.E.P., K.S.Y., M.L.K. and R.T.L. conceived and designed the study; R.T.L. coordinated data collection; C.E.P. and K.S.Y. carried out the statistical analyses; C.E.P., R.T.L. and KSY wrote the manuscript. All authors gave final approval for publication.
Competing interests. We declare we have no competing interests.
Funding. C.E.P. received funding from Trygfonden Charitable Foundation, M.L.K. received funding from the ERC.
Acknowledgements. The authors thank Amy Sewart for the assistance with data collection.

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
