## [Reviewer comments · Royal Society Open Science]

Review History

RSOS-181555.R0 (Original submission)

Review form: Reviewer 1 (Hiroshi Nittono)

Is the manuscript scientifically sound in its present form?

Yes

Are the interpretations and conclusions justified by the results?

No

Is the language acceptable?

Yes

Is it clear how to access all supporting data?

Yes

Do you have any ethical concerns with this paper?

No

Have you any concerns about statistical analyses in this paper?

Yes

Recommendation?

Major revision is needed (please make suggestions in comments)

Comments to the Author(s)

This paper reported a single experiment in which a total of 561 undergraduate students rated the valence of sound stimuli including human crying, cats' meows, and dogs' whines or whimpers. The participants were divided into four groups according to the current pet-owner status. The major findings were that those who owned pets gave more negative ratings to dog whines than those who did not own pets and that those who owned cats only gave more negative ratings to cats' distress vocalization. There was no personality difference between cat and dog owners.

I have read the paper with interest. For full disclosure, I don't like dogs or cats very much and I don't own either. In a recent online survey conducted in Japan, about 20% of Japanese people (1,000 respondents from their 20's to 60's) answered that they did not like either dogs or cats (<https://www.cross-m.co.jp/news/release/20180425.html>; in Japanese).

Nevertheless, I agree that the idea that the experience of interacting with pet animals increases the ability to detect distress vocalization of pet animals is reasonable and worth examining empirically.

I have one major concern and several minor questions and requests.

The most serious concern is why those who owned both cats and dogs did not show the tendency similar to those who owned cats alone, the latter of which ($n = 31$) the authors argued was somehow special. In my reading, I could not find the answer to this question.

A related question is why cat-alone owners had a good ability to detect distress vocalization of dogs, which they did not own and had less daily experience with. If pet owners are more sensitive to negative emotion in their pets and this ability is learned from interaction with them, the ability seems to be specific to the species that they own.

An alternative explanation is that those who own pets are originally sensitive to animals' internal states because they care for animals. This explains all the results except the cat-only owners' response to cat meows. However, as I argued above, the reason why the effect of owning cats disappeared when they had dogs at the same time is unclear. The authors may want to discuss this issue.

Minor points

1. Hypotheses. At the end of the Introduction, only a few hypotheses are listed. It would be nice to organize all the hypotheses to be tested here. In connection to this, the last sentence "We also hypothesized that cat owners would show greater anxiety and depression symptoms compared to dog owners and adults with no pets" appears suddenly. How central is this hypothesis, which is eventually rejected, to the authors' purpose in this study?
2. Sample size. Please add some comments to rationalize this sample size.
3. Stimuli. On page 6, it is unclear how many sounds were used per category. Please specify.

4. Procedure: Were headphones used for sound presentation? Please give some explanation? about how the sound intensity was adjusted.

5. Citation. On page 3, the reference number (7) of the AVMA report looks odd.

If my major question is resolved, I would be happy to support the publication of this paper.

Review form: Reviewer 2 (Céline Tallet)

Is the manuscript scientifically sound in its present form?

No

Are the interpretations and conclusions justified by the results?

No

Is the language acceptable?

Yes

Is it clear how to access all supporting data?

Yes

Do you have any ethical concerns with this paper?

No

Have you any concerns about statistical analyses in this paper?

Yes

Recommendation?

Major revision is needed (please make suggestions in comments)

Comments to the Author(s)

Dear authors,

I carefully read your article submitted to Royal Society of Open Science « Pawsitively sad : Pet-owners are more sensitive to negative emotion in animal distress vocalisations ».

The topic of the article suits well the scope of the journal, and it reports new results in the field of the cross-species interpretation of acoustic signals. However, I have concerns on its writing, and I raised questions you should answer before considering its publication.

Introduction

It is appreciated to have a societal background and the history of the topic, that explains why the question is meaningful. However, we miss information on the specificity of human perception of animal sounds. Human are able to recognize the valence and intensity of sounds from many animals, domesticated or not, so this would be nice to have a paragraph which reports this, and explain the specific interest of working on dog and cat sounds.

Also, the intensity of the emotion modifies the acoustic parameters of the sounds, and you never mention this, as you focus on valence. How can you be sure intensity did not interfere in the evaluation process?

Minor edits: 3rd paragraph, remove all the unnecessary "(7)"

5th paragraph: add a point in the end

Method

The method is not enough detailed to my mind. This is the weak part of the article. I have several questions:

- How did you select the 561 participants among the 945 of the larger study?
- How is the racial/ethnic background important for the study? If you had expectations, please report them in the introduction, and include this in the statistics.
- Did you ask people for the breed they owned? There are house cats and cats that rather spend their days outside, how could it affect the vocal expression? And thus evaluation. how long had the participants been a owner?
- Why are the ECR results not shown in Table 1 ? Is 3.96 a high score? Please be homogeneous for the three types of psycho tests. In addition you should move these data (and the corresponding text) to the results part, and probably merging table 1 and 2 would be a good option).
- We need information about the situations of the records, as you mention in the discussion there are several situations in the OxVoc database. Is the vocal expression similar/close among all the situations? "distress" may be express in a large range of situations, how did this affect the results? Did you calculate it?
- Did you use sounds from adult or young animals?
- How many sounds did people evaluate?
- How did you scale the volume? Which is highly related to intensity and valence.
- Please add a "statistical analyses" part so that we have an overview on the statistics rather than having information spread into the results

Results

- You report no gender effect on ratings, but you did not report the interaction you included in the statistical model (gender * ownership at least)
- my major concern here is that you do not know the valence of the situations: dog whines are rated as more negative than cat meows, but how is it in reality? If the situations were of different valence (excluding inter-animals variability), how can you compare evaluations from different people? I might be confused by the fact that the method lack of details, but how can you control this?
- "dogs whines are particularly distressing", this sentence, and other similar sentences, is untrue. You did not evaluate the distressing impact of the sounds. This raises the question of how you asked people to do the task. Did you ask them how distressful are the sounds? Or how negative/positive is the situation for the animal vocalizing?
- The comparison with human cries should probably be removed, as you do not explain how you did this. What it with the same people? The same volume? In the same room? The condition of the evaluation may impact the evaluation, so this seems to be very dangerous to make comparison between studies, or including results from another study.
- "no difference in cat owner's..." once again I am not convinced the use of other sounds evaluated elsewhere is suitable here.

Discussion

Do you think other parameters than the pitch may explain the differences in evaluation?

The explanation of the difference between dog and cat sounds may be sufficient, but information on the breeds you compared would improve a lot the discussion part. There is probably a huge variability between dog species, between cat species too, and not only between cats and dogs. I hope these report will help you in the process of publishing the article.

Review form: Reviewer 3 (Marek Špinka)

Is the manuscript scientifically sound in its present form?

Yes

Are the interpretations and conclusions justified by the results?

Yes

Is the language acceptable?

Yes

Is it clear how to access all supporting data?

Yes

Do you have any ethical concerns with this paper?

No

Have you any concerns about statistical analyses in this paper?

No

Recommendation?

Accept with minor revision (please list in comments)

Comments to the Author(s)

Review of Parsons et al RSOS Pawsitively sad

This is a well-designed study addressing an interesting and important question. The paper is well organized and easy to read. The results are presented in sufficient detail, including the effect sizes and Fig 1 captures the main effects in a concise way. The conclusions are firmly based in the results.

Nevertheless, I have a few comments on Introduction, on the presentation of the results and the interpretation of the results in the Discussion.

1. It is unclear why the aspects of social-dominance, competitiveness and political conservatism aspect is exposed in quite some detail in the Introduction (page 3, lines 20-31) as these aspects are then never mentioned later in the paper.

2. Page 4, lines 16-18. When referring to the Pongracz et al 2005 paper, the authors state that "(human) adults can accurately guess the context ... and correctly infer whether the dog was aggressive, fearful or playful". This is an overstatement, as the listeners in that study were above chance in guessing the context but by far not totally accurate (see Fig 6 in the cited paper), and inference about one of the three types of situations (Aggressive, fearful, playful) was correct in about 2/3 of the cases (see data in Table 5 in the cited study).

3. Page 3, lines 39-50, plus the whole section of Results and Discussion. All in all, there were five types of relationships examined in the study: A. Pet ownership <-> psychological properties; B. Psychological properties <-> sound rating; C. Pet ownership <-> sound rating; D. Species of the vocalizing animal <-> sound rating; E. acoustic properties of the calls <-> sound rating. However, this is not very transparent from the paper. For instance, only relationships A., C. and D. are mentioned in the paragraph on p 4 lines 39-50 where the aims of the study are outlined. In the Results, these five types of relationship are not always clearly identified and sometimes, they are mentioned in two separate locations in the text, e.g. the effect B on p 6 l 41-43 and on p 7 lines 17-20. Also, some of the results are missing: for instance, it is not reported whether Anxiety measure affected sound rating (effect B for GADQ). There is a need to organize this very systematically in the paper, otherwise it is difficult for the reader to keep track of questions results and interpretation. I also suggest that these 5 types of relationships are dealt with in the same order in all the sections of the manuscript (Intro, Methods, Results, Discussion), as far as possible.

4. Page 5, Table 1. How did the proportions of participants that were above the cut-off scores of EDS-5 and GAD-Q (77.4% and 88.2%) compare to the US general population? This might be important because UCLA undergraduate psychology students might differ from the general population. This applies to other variables and questions in the study. For instance, UCLA psychology students are probably politically non-conservative, so could it be that the dog-owners in this study are not really representative of the US dog owners in general? (See point 1.). Also, the participants were very young and probably lived in small households. Age and household size may influence some of the variables examined in the study substantially, e.g. (1, 2). Would the results be the same with people of different age and social strata? I do not propose that the authors engage on extensive speculations about what other factors that were not investigated might have effects on psychological profiles of dog and cat owners. Nevertheless, it would be appropriate to mention in the discussion that the participants in the study were a particular subset of the human population and that dog / cat owners / non-owners from different strata of the population might have different characteristics and the results of the current study may not be generalizable to them, and mention specifically some of the possible differences, based on the available data from other studies.

5. It is very good that effect sizes are reported for all the results. Nevertheless, there seems to be a certain bias in interpreting these effect sizes. Specifically, a lot of attention in the paper is devoted to the interaction between pet-owning status and animal species being rated (e.g., abstract lines 16-17, first paragraph of the discussion, Conclusion). However, the effect size for this interaction is very small ($\eta^2 = 0.01$, p 7 lines 32-33) and is 50 times smaller than the effect size of the animal being rated ($\eta^2 = 0.51$, p 7 lines 30-31). Also the pet-owning status effect is small ($\eta^2 = 0.04$, p 7 lines 31-32). For other types of effect, the small size of the effect is clearly stated and discussed (e.g., p. 3 lines 21-22, p. 11 lines 14-15) but for "main" results of the study, the authors are silent about their small effect sizes. I agreed it is interesting that cat-only owners were more sensitive to cat meows than other groups but it needs to be conveyed clearly and prominently to the reader that this effect explained only 1% of the total variability in the sound rating.

6. Cat-and-dog owners seem to be similar to dog-only owners, but less so to cat-only owners. This result is not discussed in the paper.

7. Page 10 lines 33-35. This explanation lacks internal logic. The fact that ALL participants found the dog whines particularly plaintive cannot be explained by the fact that DOG OWNERS have a greater tendency to anthropomorphize.

8. I found the raw data, but not the code of the statistical evaluation, on the data depository. In conclusion, I recommend a minor revision of the paper before it will be accepted for publication in RSOS.

References

1. Murray JK, Browne WJ, Roberts MA, Whitmarsh A, Gruffydd-Jones TJ. Number and ownership profiles of cats and dogs in the UK. *Veterinary Record*. 2010;166(6):163-8.
2. Jorm AF. Does old age reduce the risk of anxiety and depression? A review of epidemiological studies across the adult life span. *Psychological Medicine*. 2000;30(1):11-22.

Decision letter (RSOS-181555.R0)

12-Feb-2019

Dear Dr Parsons,

The editors assigned to your paper ("Pawsitively sad: Pet-owners are more sensitive to negative

emotion in animal distress vocalisations") have now received comments from reviewers. We would like you to revise your paper in accordance with the referee and Associate Editor suggestions which can be found below (not including confidential reports to the Editor). Please note this decision does not guarantee eventual acceptance.

Please submit a copy of your revised paper before 07-Mar-2019. Please note that the revision deadline will expire at 00.00am on this date. If we do not hear from you within this time then it will be assumed that the paper has been withdrawn. In exceptional circumstances, extensions may be possible if agreed with the Editorial Office in advance. We do not allow multiple rounds of revision so we urge you to make every effort to fully address all of the comments at this stage. If deemed necessary by the Editors, your manuscript will be sent back to one or more of the original reviewers for assessment. If the original reviewers are not available, we may invite new reviewers.

- Data accessibility

If you wish to submit your supporting data or code to Dryad (<http://datadryad.org/>), or modify your current submission to dryad, please use the following link:
<http://datadryad.org/submit?journalID=RSOS&manu=RSOS-181555>

- Competing interests

- Authors' contributions

- Acknowledgements

- Funding statement

on behalf of Dr Anastasia Christakou (Associate Editor) and Antonia Hamilton (Subject Editor)
openscience@royalsociety.org

Comments to Author:

Reviewers' Comments to Author:

Reviewer: 1

Comments to the Author(s)

This paper reported a single experiment in which a total of 561 undergraduate students rated the valence of sound stimuli including human crying, cats' meows, and dogs' whines or whimpers. The participants were divided into four groups according to the current pet-owner status. The major findings were that those who owned pets gave more negative ratings to dog whines than those who did not own pets and that those who owned cats only gave more negative ratings to cats' distress vocalization. There was no personality difference between cat and dog owners.

I have read the paper with interest. For full disclosure, I don't like dogs or cats very much and I

don't own either. In a recent online survey conducted in Japan, about 20% of Japanese people (1,000 respondents from their 20's to 60's) answered that they did not like either dogs or cats (<https://www.cross-m.co.jp/news/release/20180425.html>; in Japanese).

Nevertheless, I agree that the idea that the experience of interacting with pet animals increases the ability to detect distress vocalization of pet animals is reasonable and worth examining empirically.

I have one major concern and several minor questions and requests.

The most serious concern is why those who owned both cats and dogs did not show the tendency similar to those who owned cats alone, the latter of which (n = 31) the authors argued was somehow special. In my reading, I could not find the answer to this question.

A related question is why cat-alone owners had a good ability to detect distress vocalization of dogs, which they did not own and had less daily experience with. If pet owners are more sensitive to negative emotion in their pets and this ability is learned from interaction with them, the ability seems to be specific to the species that they own.

An alternative explanation is that those who own pets are originally sensitive to animals' internal states because they care for animals. This explains all the results except the cat-only owners' response to cat meows. However, as I argued above, the reason why the effect of owning cats disappeared when they had dogs at the same time is unclear. The authors may want to discuss this issue.

Minor points

1. Hypotheses. At the end of the Introduction, only a few hypotheses are listed. It would be nice to organize all the hypotheses to be tested here. In connection to this, the last sentence "We also hypothesized that cat owners would show greater anxiety and depression symptoms compared to dog owners and adults with no pets" appears suddenly. How central is this hypothesis, which is eventually rejected, to the authors' purpose in this study?
2. Sample size. Please add some comments to rationalize this sample size.
3. Stimuli. On page 6, it is unclear how many sounds were used per category. Please specify.
4. Procedure: Were headphones used for sound presentation? Please give some explanation about how the sound intensity was adjusted.
5. Citation. On page 3, the reference number (7) of the AVMA report looks odd.

If my major question is resolved, I would be happy to support the publication of this paper.

Reviewer: 2

Comments to the Author(s)

Dear authors,

I carefully read your article submitted to Royal Society of Open Science « Pawsitively sad : Pet-owners are more sensitive to negative emotion in animal distress vocalisations ».

The topic of the article suits well the scope of the journal, and it reports new results in the field of the cross-species interpretation of acoustic signals. However, I have concerns on its writing, and I raised questions you should answer before considering its publication.

Introduction

It is appreciated to have a societal background and the history of the topic, that explains why the question is meaningful. However, we miss information on the specificity of human perception of animal sounds. Humans are able to recognize the valence and intensity of sounds from many animals, domesticated or not, so this would be nice to have a paragraph which reports this, and explain the specific interest of working on dog and cat sounds.

Also, the intensity of the emotion modifies the acoustic parameters of the sounds, and you never mention this, as you focus on valence. How can you be sure intensity did not interfere in the evaluation process?

Minor edits: 3rd paragraph, remove all the unnecessary "(7)"

5th paragraph: add a point in the end

Method

The method is not enough detailed to my mind. This is the weak part of the article. I have several questions:

- How did you select the 561 participants among the 945 of the larger study?
- How is the racial/ethnic background important for the study? If you had expectations, please report them in the introduction, and include this in the statistics.
- Did you ask people for the breed they owned? There are house cats and cats that rather spend their days outside, how could it affect the vocal expression? And thus evaluation. how long had the participants been a owner?
- Why are the ECR results not shown in Table 1? Is 3.96 a high score? Please be homogeneous for the three types of psycho tests. In addition you should move these data (and the corresponding text) to the results part, and probably merging table 1 and 2 would be a good option).
- We need information about the situations of the records, as you mention in the discussion there are several situations in the OxVoc database. Is the vocal expression similar/close among all the situations? "distress" may be expressed in a large range of situations, how did this affect the results? Did you calculate it?
- Did you use sounds from adult or young animals?
- How many sounds did people evaluate?
- How did you scale the volume? Which is highly related to intensity and valence.
- Please add a "statistical analyses" part so that we have an overview on the statistics rather than having information spread into the results

Results

- You report no gender effect on ratings, but you did not report the interaction you included in the statistical model (gender * ownership at least)
- my major concern here is that you do not know the valence of the situations: dog whines are rated as more negative than cat meows, but how is it in reality? If the situations were of different valence (excluding inter-animals variability), how can you compare evaluations from different people? I might be confused by the fact that the method lack of details, but how can you control this?
- "dogs whines are particularly distressing", this sentence, and other similar sentences, is untrue. You did not evaluate the distressing impact of the sounds. This raises the question of how you asked people to do the task. Did you ask them how distressful are the sounds? Or how negative/positive is the situation for the animal vocalizing?
- The comparison with human cries should probably be removed, as you do not explain how you did this. What it with the same people? The same volume? In the same room? The condition of

the evaluation may impact the evaluation, so this seems to be very dangerous to make comparison between studies, or including results from another study.

- “no difference in cat owner’s...” once again I am not convinced the use of other sounds evaluated elsewhere is suitable here.

Discussion

Do you think other parameters than the pitch may explain the differences in evaluation?

The explanation of the difference between dog and cat sounds may be sufficient, but information on the breeds you compared would improve a lot the discussion part. There is probably a huge variability between dog species, between cat species too, and not only between cats and dogs.

I hope these report will help you in the process of publishing the article.

Reviewer: 3

Comments to the Author(s)

Review of Parsons et al RSOS Pawsitively sad

This is a well-designed study addressing an interesting and important question. The paper is well organized and easy to read. The results are presented in sufficient detail, including the effect sizes and Fig 1 captures the main effects in a concise way. The conclusions are firmly based in the results.

Nevertheless, I have a few comments on Introduction, on the presentation of the results and the interpretation of the results in the Discussion.

1. It is unclear why the aspects of social-dominance, competitiveness and political conservatism aspect is exposed in quite some detail in the Introduction (page 3, lines 20-31) as these aspects are then never mentioned later in the paper.

2. Page 4, lines 16-18. When referring to the Pongracz et al 2005 paper, the authors state that “(human) adults can accurately guess the context ... and correctly infer whether the dog was aggressive, fearful or playful”. This is an overstatement, as the listeners in that study were above chance in guessing the context but by far not totally accurate (see Fig 6 in the cited paper), and inference about one of the three types of situations (Aggressive, fearful, playful) was correct in about 2/3 of the cases (see data in Table 5 in the cited study).

3. Page 3, lines 39-50, plus the whole section of Results and Discussion. All in all, there were five types of relationships examined in the study: A. Pet ownership <-> psychological properties; B. Psychological properties <-> sound rating; C. Pet ownership <-> sound rating; D. Species of the vocalizing animal <-> sound rating; E. acoustic properties of the calls <-> sound rating. However, this is not very transparent from the paper. For instance, only relationships A., C. and D. are mentioned in the paragraph on p 4 lines 39-50 where the aims of the study are outlined. In the Results, these five types of relationship are not always clearly identified and sometimes, they are mentioned in two separate locations in the text, e.g. the effect B on p 6 l 41-43 and on p 7 lines 17-20. Also, some of the results are missing: for instance, it is not reported whether Anxiety measure affected sound rating (effect B for GADQ). There is a need to organize this very systematically in the paper, otherwise it is difficult for the reader to keep track of questions results and interpretation. I also suggest that these 5 types of relationships are dealt with in the same order in all the sections of the manuscript (Intro, Methods, Results, Discussion), as far as possible.

4. Page 5, Table 1. How did the proportions of participants that were above the cut-off scores of EDS-5 and GAD-Q (77.4% and 88.2%) compare to the US general population? This might be important because UCLA undergraduate psychology students might differ from the general population. This applies to other variables and questions in the study. For instance, UCLA psychology students are probably politically non-conservative, so could it be that the dog-owners in this study are not really representative of the US dog owners in general? (See point 1.). Also, the participants were very young and probably lived in small households. Age and household

size may influence some of the variables examined in the study substantially, e.g. (1, 2). Would the results be the same with people of different age and social strata? I do not propose that the authors engage on extensive speculations about what other factors that were not investigated might have effects on psychological profiles of dog and cat owners. Nevertheless, it would be appropriate to mention in the discussion that the participants in the study were a particular subset of the human population and that dog / cat owners / non-owners from different strata of the population might have different characteristics and the results of the current study may not be generalizable to them, and mention specifically some of the possible differences, based on the available data from other studies.

5. It is very good that effect sizes are reported for all the results. Nevertheless, there seems to be a certain bias in interpreting these effect sizes. Specifically, a lot of attention in the paper is devoted to the interaction between pet-owning status and animal species being rated (e.g., abstract lines 16-17, first paragraph of the discussion, Conclusion). However, the effect size for this interaction is very small (eta squared = 0.01, p 7 lines 32-33) and is 50 times smaller than the effect size of the animal being rated (eta squared = 0.51, p 7 lines 30-31). Also the pet-owning status effect is small (eta squared = 0.04, p 7 lines 31-32). For other types of effect, the small size of the effect is clearly stated and discussed (e.g., p. 3 lines 21-22, p. 11 lines 14-15) but for "main" results of the study, the authors are silent about their small effect sizes. I agreed it is interesting that cat-only owners were more sensitive to cat meows than other groups but it needs to be conveyed clearly and prominently to the reader that this effect explained only 1% of the total variability in the sound rating.

6. Cat-and-dog owners seem to be similar to dog-only owners, but less so to cat-only owners. This result is not discussed in the paper.

7. Page 10 lines 33-35. This explanation lacks internal logic. The fact that ALL participants found the dog whines particularly plaintive cannot be explained by the fact that DOG OWNERS have a greater tendency to anthropomorphize.

8. I found the raw data, but not the code of the statistical evaluation, on the data depository. In conclusion, I recommend a minor revision of the paper before it will be accepted for publication in RSOS.

References

1. Murray JK, Browne WJ, Roberts MA, Whitmarsh A, Gruffydd-Jones TJ. Number and ownership profiles of cats and dogs in the UK. *Veterinary Record*. 2010;166(6):163-8.
2. Jorm AF. Does old age reduce the risk of anxiety and depression? A review of epidemiological studies across the adult life span. *Psychological Medicine*. 2000;30(1):11-22.

Author's Response to Decision Letter for (RSOS-181555.R0)

See Appendices A & B.

RSOS-181555.R1 (Revision)

Review form: Reviewer 1 (Hiroshi Nittono)

Is the manuscript scientifically sound in its present form?

Yes

Are the interpretations and conclusions justified by the results?

Yes

Is the language acceptable?

Yes

Is it clear how to access all supporting data?

Yes

Do you have any ethical concerns with this paper?

No

Have you any concerns about statistical analyses in this paper?

No

Recommendation?

Accept with minor revision (please list in comments)

Comments to the Author(s)

The authors did a good job in reorganizing the manuscript and reaching a more balanced conclusion.

I still have some concern about the specificity of the cat-only owner group (n = 31, the smallest group), but the current finding may provide a hint for future studies.

Although it may not be directly related to the authors' interest, I think it would be informative to show the data about the human baby and adult cryings more concretely. Namely, physical parameters (Table 2) and pirate plots (Figure 2) of these sound types. If the space is limited, they can be supplementary materials.

Minor suggestion: It would be nice to keep the number of digits (places) constant when stats are reported.

Review form: Reviewer 2 (Céline Tallet)

Is the manuscript scientifically sound in its present form?

Yes

Are the interpretations and conclusions justified by the results?

Yes

Is the language acceptable?

Yes

Is it clear how to access all supporting data?

Yes

Do you have any ethical concerns with this paper?

No

Have you any concerns about statistical analyses in this paper?

No

Recommendation?

Accept as is

Comments to the Author(s)

Dear authors,

Your article has been greatly improved by your answers to the reviewers, and it is now very pleasant to read it. I appreciate the additional online survey you did, and all the efforts to revise the text.

Review form: Reviewer 3 (Marek Špinka)

Is the manuscript scientifically sound in its present form?

Yes

Are the interpretations and conclusions justified by the results?

Yes

Is the language acceptable?

Yes

Is it clear how to access all supporting data?

Yes

Do you have any ethical concerns with this paper?

No

Have you any concerns about statistical analyses in this paper?

No

Recommendation?

Accept as is

Comments to the Author(s)

The authors dealt with the weaknesses of the original submission very well with the revision. In my view, the paper is ready of publication in RSOS. It was a pleasure to review this interesting study.

Decision letter (RSOS-181555.R1)

14-Jun-2019

Dear Dr Parsons:

On behalf of the Editors, I am pleased to inform you that your Manuscript RSOS-181555.R1 entitled "Pawsitively sad: Pet-owners are more sensitive to negative emotion in animal distress

vocalisations" has been accepted for publication in Royal Society Open Science subject to minor revision in accordance with the referee suggestions. Please find the referees' comments at the end of this email.

The reviewers and Subject Editor have recommended publication, but also suggest some minor revisions to your manuscript. Therefore, I invite you to respond to the comments and revise your manuscript.

- Ethics statement

- Data accessibility

If you wish to submit your supporting data or code to Dryad (<http://datadryad.org/>), or modify your current submission to dryad, please use the following link:
<http://datadryad.org/submit?journalID=RSOS&manu=RSOS-181555.R1>

- Competing interests

- Authors' contributions

- Acknowledgements

- Funding statement

Because the schedule for publication is very tight, it is a condition of publication that you submit the revised version of your manuscript before 23-Jun-2019. Please note that the revision deadline will expire at 00.00am on this date. If you do not think you will be able to meet this date please let me know immediately.

on behalf of Dr Anastasia Christakou (Associate Editor) and Antonia Hamilton (Subject Editor)
openscience@royalsociety.org

Reviewer comments to Author:
Reviewer: 2

Comments to the Author(s)

Dear authors,
Your article has been greatly improved by your answers to the reviewers, and it is now very pleasant to read it. I appreciate the additional online survey you did, and all the efforts to revise the text.

Reviewer: 1

Comments to the Author(s)

The authors did a good job in reorganizing the manuscript and reaching a more balanced conclusion.

I still have some concern about the specificity of the cat-only owner group ($n = 31$, the smallest group), but the current finding may provide a hint for future studies.

Although it may not be directly related to the authors' interest, I think it would be informative to show the data about the human baby and adult cryings more concretely. Namely, physical parameters (Table 2) and pirate plots (Figure 2) of these sound types. If the space is limited, they can be supplementary materials.

Minor suggestion: It would be nice to keep the number of digits (places) constant when stats are reported.

Reviewer: 3

Comments to the Author(s)

The authors dealt with the weaknesses of the original submission very well with the revision. In my view, the paper is ready of publication in RSOS. It was a pleasure to review this interesting study.

Author's Response to Decision Letter for (RSOS-181555.R1)

See Appendix C.

Decision letter (RSOS-181555.R2)

09-Jul-2019

Dear Dr Parsons,

I am pleased to inform you that your manuscript entitled "Pawsitively sad: Pet-owners are more sensitive to negative emotion in animal distress vocalisations" is now accepted for publication in Royal Society Open Science.

Kind regards,

on behalf of Dr Anastasia Christakou (Associate Editor) and Antonia Hamilton (Subject Editor)
openscience@royalsociety.org

Appendix A

AARHUS UNIVERSITY

Manuscript ID RSOS-181555

"Pawsitively sad: Pet-owners are more sensitive to negative emotion in animal distress vocalisations"

Dear Editor,

We thank the reviewers for their comments, 'I have read the paper with interest ...and I agree that the idea that the experience of interacting with pet animals increases the ability to detect distress vocalization of pet animals is reasonable and worth examining empirically' and 'The topic of the article suits well the scope of the journal, and it reports new results in the field of the cross-species interpretation of acoustic signals' and 'This is a well-designed study addressing an interesting and important question. The paper is well organized and easy to read. The results are presented in sufficient detail, including the effect sizes and Fig 1 captures the main effects in a concise way. The conclusions are firmly based in the results'.

These reviews have provided important suggestions for improving our manuscript. We have substantially revised our manuscript and present a point-by-point response to each comment received below. In brief, we carried out an additional online experiment, in order to respond fully to two of the reviewers. We have also re-organised the manuscript as suggested, and provided additional reflections on the large population tested here.

We have made every effort to fully address the review comments, and hope the manuscript is now ready for publication.

Yours sincerely,

Christine Parsons (on behalf of the authors)

Appendix B

Manuscript ID RSOS-181555

“Pawsitively sad: Pet-owners are more sensitive to negative emotion in animal distress vocalisations”

Reviewer 1

I have one major concern and several minor questions and requests. The most serious concern is why those who owned both cats and dogs did not show the tendency similar to those who owned cats alone, the latter of which ($n = 31$) the authors argued was somehow special. In my reading, I could not find the answer to this question.

We thank the reviewer for raising this important point, which we should have given more consideration to in discussing the results. We have added a paragraph to the discussion to address this. We speculate that if an adult has both a cat and dog, the dog is likely to be the dominant pet, requiring more caregiving (walks, more frequent trips outside). We speculate that having a dog and a cat means that the individual may be more attuned to the dog, and less to the cat. In this way, we argue that people who chose a cat alone are distinct from those who own both. More broadly, we emphasise throughout that dogs are more familiar, and more woven into the fabric of our lives than cats. There are numerous analyses to show that we seek out dog-related content more frequently on the internet (Google Searches, video-searches, Instagram focused accounts) (1). Even where we are not seeking out animal-related material, dogs are seen in TV commercials more frequently than cats (2).

A related question is why cat-alone owners had a good ability to detect distress vocalization of dogs, which they did not own and had less daily experience with. If pet owners are more sensitive to negative emotion in their pets and this ability is learned from interaction with them, the ability seems to be specific to the species that they own. An alternative explanation is that those who own pets are originally sensitive to animals' internal states because they care for animals. This explains all the results except the cat-only owners' response to cat meows. However, as I argued above, the reason why the effect of owning cats disappeared when they had dogs at the same time is unclear. The authors may want to discuss this issue.

This is an excellent point. We have added this interpretation to the discussion (conclusion) section. This is a ‘chicken and egg’ point, so we have also removed points about experience-dependent effects elsewhere in the manuscript (e.g., abstract).

Minor points

1. Hypotheses. At the end of the Introduction, only a few hypotheses are listed. It would be nice to organize all the hypotheses to be tested here. In connection to this, the last sentence "We also hypothesized that cat owners would show greater anxiety and depression symptoms compared to dog owners and adults with no pets" appears suddenly. How central is this hypothesis, which is eventually rejected, to the authors' purpose in this study?

This is a secondary hypothesis – but given the previous findings on personality differences, we felt it was worth testing this, especially given the sample size available to us. We have re-organised the hypotheses in line with another reviewer’s suggestion and hope this is less abrupt now.

2. Sample size. Please add some comments to rationalize this sample size.

We used this large sample size because we were performing a validation study of a sounds database. We wanted to ensure a sufficiently large sample to be able to test the reliability of ratings and categorisation of sounds in this database. We have added details on this as suggested (“These participants were drawn from a

larger study of 945 adults who took part in a validation study for the OxVoc (the sample provided adequate power to examine test-retest reliability and categorisation validity; Oxford Vocal Sounds Database)”.

3. Stimuli. On page 6, it is unclear how many sounds were used per category. Please specify.

We have added this information (30 stimuli for the domestic animal sounds).

4. Procedure: Were headphones used for sound presentation? Please give some explanation about how the sound intensity was adjusted.

We have added information on the headphones used.

5. Citation. On page 3, the reference number (7) of the AVMA report looks odd.

There was an error in our reference manager and we have now corrected this.

Reviewer 2.

We miss information on the specificity of human perception of animal sounds. Humans are able to recognize the valence and intensity of sounds from many animals, domesticated or not, so this would be nice to have a paragraph which reports this, and explain the specific interest of working on dog and cat sounds.

We agree with the reviewer that humans are able to recognise the valence and intensity of sounds from many animals. We have now added half a paragraph to the start of the introduction to better frame why we are interested in cats and dogs specifically – as these are our most commonly chosen domestic companions.

Also, the intensity of the emotion modifies the acoustic parameters of the sounds, and you never mention this, as you focus on valence. How can you be sure intensity did not interfere in the evaluation process?

We agree with the reviewer here. We did have a procedure in place, described in our previous article, to match the intensity of the stimuli. We have added information on this, and thank the reviewer for raising this.

Minor edits: 3rd paragraph, remove all the unnecessary “(7)”

This has been amended.

5th paragraph: add a point in the end

Amended.

The method is not enough detailed to my mind. This is the weak part of the article. I have several questions:

We are grateful for the opportunity to address the lack of detail here. We elaborate on all of the suggested points.

- *How did you select the 561 participants among the 945 of the larger study*

This was the first 561 recruited to the study. We tested the first group of participants with the animal sounds, and the remaining participants with musical stimuli. We have added this information to the manuscript.

- *How is the racial/ethnic background important for the study? If you had expectations, please report them in the introduction, and include this in the statistics.*

We did not have any specific expectations in relation to the racial/ethnic background of participants. Rather, we report this to provide the reader with details on the student population of UCLA, which we note is not representative of the general US population. We have added more information to our limitations section on the potential biases within this sample.

• *Did you ask people for the breed they owned? There are house cats and cats that rather spend their days outside, how could it affect the vocal expression? And thus evaluation. How long had the participants been a owner?*

We did not collect this information. We have added this as a limitation in the discussion.

• *Why are the ECR results not shown in Table 1 ? Is 3.96 a high score? Please be homogeneous for the three types of psycho tests. In addition you should move these data (and the corresponding text) to the results part, and probably merging table 1 and 2 would be a good option).*

We have removed Table 1 and moved all other findings, as the reviewer suggested. We also note that the ECR is a measure of attachment style, and unlike the GAD-Q or EDS it does not have a typical cut-off score. Instead we present more information on the scoring of the ECR, to increase the interpretability of the results.

• *We need information about the situations of the records, as you mention in the discussion there are several situations in the OxVoc database. Is the vocal expression similar/close among all the situations? “distress” may be express in a large range of situations, how did this affect the results? Did you calculate it?*

We do not know about the actual situations that generated the vocalisations from the animals. To counter this limitation, we recruited 31 participants via an online survey platform (Prolific; an online crowd-working subject pool). Each participant listened to the cat and dog vocalisations and categorised the stimulus as either ‘cat’ or ‘kitten’ or ‘puppy’ or ‘dog’, and also the ‘situation’ of the sound, as the reviewer refers to it. We asked participants to categorise if the animal sounded ‘hungry’, ‘in pain’, ‘seeking general attention’ or if they ‘cannot say’. The order of these two questions was randomised across participants, and the order of the stimulus presentation was also randomised across participants. Participant inclusion criteria was: aged between 18- 60 years (inclusive) and no self-reported hearing difficulties. The study included an attention check, where participants had to select the type of vocalisation they had heard (e.g., meow or purr).

We have added the data from this ‘stimulus check’ experiment to our manuscript and supplementary materials. In brief, we found no difference between the numbers of cat and dog stimuli categorised as adult or infant, or in the ‘reason’ for the vocalisation.

• *Did you use sounds from adult or young animals?*

See response above.

• *How many sounds did people evaluate?*

We have added this information (animal sounds n = 30, human baby cry sounds n = 21, female adult cry sounds n = 21).

• *How did you scale the volume? Which is highly related to intensity and valence.*

We have added information on the between-stimulus intensity matching procedures we followed to ensure that volume was comparable across different sounds.

• Please add a “statistical analyses” part so that we have an overview on the statistics rather than having information spread into the results

We have added this section as requested.

Results

*• You report no gender effect on ratings, but you did not report the interaction you included in the statistical model (gender * ownership at least)*

We found no main effect of gender, so we did not include it as a factor in our models. We have moved this to our section on Statistical analysis and clarified.

• my major concern here is that you do not know the valence of the situations: dog whines are rated as more negative than cat meows, but how is it in reality? If the situations were of different valence (excluding inter-animals variability), how can you compare evaluations from different people? I might be confused by the fact that the method lack of details, but how can you control this?

We do not have any data on the situations that these sounds arose in. Our initial selection of the sounds was with the aim of obtaining clear exemplars of cat and dog distress vocalisations. This is a limitation, but we made considerable efforts to address this limitation, via additional data collection. See Response to Reviewer 1, for further comment. Finally, we are comparing evaluations of different sounds from the same people.

• “dogs whines are particularly distressing”, this sentence, and other similar sentences, is untrue. You did not evaluate the distressing impact of the sounds. This raises the question of how you asked people to do the task. Did you ask them how distressful are the sounds? Or how negative/positive is the situation for the animal vocalizing?

We have corrected this to ‘dog whines are perceived as especially sad’. We asked the participants to rate the emotional content of the sounds on a scale from ‘very happy to very sad’. We acknowledge that that there is a distinction between how the sound makes the listener feel (how distressing it is) and the perceived emotional content.

• The comparison with human cries should probably be removed, as you do not explain how you did this. What it with the same people? The same volume? In the same room? The condition of the evaluation may impact the evaluation, so this seems to be very dangerous to make comparison between studies, or including results from another study.

We apologise that this was unclear to the reviewer, and we have amended the method section. We played the same participants 21 baby cry sounds and 19 adult cry sounds during the same testing period, under exactly the same conditions, at the same volume, in the same room.

• “no difference in cat owner’s...” once again I am not convinced the use of other sounds evaluated elsewhere is suitable here.

As above, we have now clarified that all sounds were evaluated at the same time, in the same participant group.

Discussion

Do you think other parameters than the pitch may explain the differences in evaluation?

There may be additional acoustic parameters that contribute to participants' differences in evaluation, including the number of silences within the vocalisations (3), roughness (4) a higher frequency range (5). However, there is evidence that humans rely on the pitch cues to rate the distress vocalisations of other species with reasonable accuracy (e.g. piglet's calls, (6); dogs' calls, (7); cats' solicitation purrs, (8)). In fact, this phenomenon has been referred to as the 'pitch rule' (5). Based on this, we believe that it is reasonable to focus our analyses on pitch, although we note that there are additional parameters of interest in future work (particularly roughness and within-utterance silences).

The explanation of the difference between dog and cat sounds may be sufficient, but information on the breeds you compared would improve a lot the discussion part. There is probably a huge variability between dog species, between cat species too, and not only between cats and dogs.

We agree with the reviewer that there may be differences between breeds of cats and dogs (we assume the reviewer means breeds here). However, we believe our findings serve as an interesting start point for further work. For instance, we might hypothesise that breeds with different roles (e.g., dogs with 'guard' functions) might have distinct distress vocalisations from other breeds (e.g., hunting dogs), not least because domestic dogs vary considerably in their body size. This is in contrast to domestic cats, who are arguably more similar in their body size.

Reviewer 3.

1. It is unclear why the aspects of social-dominance, competitiveness and political conservatism aspect is exposed in quite some detail in the Introduction (page 3, lines 20-31) as these aspects are then never mentioned later in the paper.

This is a fair point, and we now interleave this into our discussion of the participant group used (who are likely to be more liberal than the general US population, although we did not collect information on political orientation).

2. Page 4, lines 16-18. When referring to the Pongracz et al 2005 paper, the authors state that "(human) adults can accurately guess the context ... and correctly infer whether the dog was aggressive, fearful or playful". This is an overstatement, as the listeners in that study were above chance in guessing the context but by far not totally accurate (see Fig 6 in the cited paper), and inference about one of the three types of situations (Aggressive, fearful, playful) was correct in about 2/3 of the cases (see data in Table 5 in the cited study).

We have amended this statement to more precisely describe the Pongracz study findings 'For instance, adults can guess, with reasonable accuracy, the context in which a dog bark occurs, scoring above chance in one study (40% accuracy rate in a 6 category choice task (9)).' We thank the reviewer for pointing this out.

3. Page 3, lines 39-50, plus the whole section of Results and Discussion. All in all, there were five types of relationships examined in the study: A. Pet ownership <-> psychological properties; B. Psychological properties <-> sound rating; C. Pet ownership <-> sound rating; D. Species of the vocalizing animal <-> sound rating; E. acoustic properties of the calls <-> sound rating. However, this is not very transparent from the paper. For instance, only relationships A., C. and D. are mentioned in the paragraph on p 4 lines 39-50 where the aims of the study are outlined.

In the Results, these five types of relationship are not always clearly identified and sometimes, they are mentioned in two separate locations in the text, e.g. the effect B on p 6 l 41-43 and on p 7 lines 17-20. Also, some of the results are missing: for instance, it is not reported whether Anxiety measure affected sound rating (effect B for GADQ). There is a need to organize this very systematically in the paper, otherwise it is difficult for the reader to keep track of questions results and interpretation. I also suggest that these 5 types of

relationships are dealt with in the same order in all the sections of the manuscript (Intro, Methods, Results, Discussion), as far as possible.

We have re-examined the layout of our paper and have restructured and added ‘signposting’ in response to the reviewer’s comment. We present two analyses: 1. We examine the differences between pet-owners and non-pet owners on our 3 psychological measures (ECR, GAD-Q and EDS). We consider the effect of B (Psychological properties <-> sound rating) to be secondary, and we note this. We have also added the GAD-Q – dog and cat vocalisation ratings correlation (non-significant). 2. A 2 x 2 design: PET OWNERSHIP (owner, not) X ANIMAL (cat, dog), with the dependent Variable of ‘sound rating’. All of our analyses are structured in this way, and we have re-ordered our discussion so it follows the results section.

4. Page 5, Table 1. How did the proportions of participants that were above the cut-off scores of EDS-5 and GAD-Q (77.4% and 88.2%) compare to the US general population? This might be important because UCLA undergraduate psychology students might differ from the general population. This applies to other variables and questions in the study. For instance, UCLA psychology students are probably politically non-conservative, so could it be that the dog-owners in this study are not really representative of the US dog owners in general? (See point 1.). Also, the participants were very young and probably lived in small households. Age and household size may influence some of the variables examined in the study substantially, e.g. (1, 2). Would the results be the same with people of different age and social strata? I do not propose that the authors engage on extensive speculations about what other factors that were not investigated might have effects on psychological profiles of dog and cat owners. Nevertheless, it would be appropriate to mention in the discussion that the participants in the study were a particular subset of the human population and that dog / cat owners / non-owners from different strata of the population might have different characteristics and the results of the current study may not be generalizable to them, and mention specifically some of the possible differences, based on the available data from other studies.

We understand the reviewer’s point about the population tested being a highly-selective and potentially non-representative university cohort. We now have a full paragraph on this in the ‘limitations’ section and we have added more on the political orientation (which we did not measure) of the students. As for the depression and anxiety scores, meta-analytic work suggests that depression is higher in university populations than in the general population (10), and anxiety-distress may also be elevated in university students compared to an age-matched group (11). Finally, we note that the average GAD-Q scores reported here were similar to other university samples (e.g., university students in Houston (12)).

5. It is very good that effect sizes are reported for all the results. Nevertheless, there seems to be a certain bias in interpreting these effect sizes. Specifically, a lot of attention in the paper is devoted to the interaction between pet-owning status and animal species being rated (e.g., abstract lines 16-17, first paragraph of the discussion, Conclusion). However, the effect size for this interaction is very small (eta squared = 0.01, p 7 lines 32-33) and is 50 times smaller than the effect size of the animal being rated (eta squared = 0.51, p 7 lines 30-31). Also the pet-owning status effect is small (eta squared = 0.04, p 7 lines 31-32). For other types of effect, the small size of the effect is clearly stated and discussed (e.g., p. 3 lines 21-22, p. 11 lines 14-15) but for “main” results of the study, the authors are silent about their small effect sizes. I agreed it is interesting that cat-only owners were more sensitive to cat meows than other groups but it needs to be conveyed clearly and prominently to the reader that this effect explained only 1% of the total variability in the sound rating.

This is a fair point, and we have now clarified that the ‘cat owner only’ finding is small in magnitude. We do this in multiple places throughout the manuscript. We clearly specify throughout which effects are small/subtle and which are larger.

6. Cat-and-dog owners seem to be similar to dog-only owners, but less so to cat-only owners. This result is not discussed in the paper.

See Response to Reviewer 1, point 1.

7. Page 10 lines 33-35. *This explanation lacks internal logic. The fact that ALL participants found the dog whines particularly plaintive cannot be explained by the fact that DOG OWNERS have a greater tendency to anthropomorphize.*

We have changed this sentence. We could not find any studies on non-dog owners and greater anthropomorphizing of dogs over cats. We clarify that if this effect is a general one, it might account for our findings.

8. *I found the raw data, but not the code of the statistical evaluation, on the data depository.*

We have added the syntax for our statistical tests on the data repository.

References

1. Philips O. The numbers don't lie: Dogs are the internet's favorite animal. The Outline 2018.
2. Williams J. Cats in Commercials. 2013.
3. Anderson A. <https://chatterbaby.org/pages/faq> [
4. Koutseff A, Reby D, Martin O, Levrero F, Patural H, Mathevon N. The acoustic space of pain: Cries as indicators of distress recovering dynamics in pre-verbal infants. *Bioacoustics*. 2018;27(4):313-25.
5. Kelly T, Mathevon N, Reby D, Levréro F, Keenan S, Gustafsson E, et al. Adult human perception of distress in the cries of bonobo, chimpanzee, and human infants. *Biological Journal of the Linnean Society*. 2017;120(4):919-30.
6. Maruščáková IL, Linhart P, Ratcliffe VF, Tallet C, Reby D, Špinka M. Humans (*Homo sapiens*) judge the emotional content of piglet (*Sus scrofa domestica*) calls based on simple acoustic parameters, not personality, empathy, nor attitude toward animals. *Journal of Comparative Psychology*. 2015;129(2):121-31.
7. Faragó T, Andics A, Devecseri V, Kis A, Gácsi M, Miklósi A. Humans rely on the same rules to assess emotional valence and intensity in conspecific and dog vocalizations. *Biology Letters*. 2014;10(1).
8. McComb K, Taylor AM, Wilson C, Charlton BD. The cry embedded within the purr. *Current Biology*. 2009;19(13):R507-R8.
9. Pongrácz P, Molnár C, Miklósi Á, Csányi V. Human listeners are able to classify dog (*Canis familiaris*) barks recorded in different situations. *Journal of Comparative Psychology*. 2005;119(2):136-44.
10. Ibrahim AK, Kelly SJ, Adams CE, Glazebrook C. A systematic review of studies of depression prevalence in university students. *Journal of psychiatric research*. 2013;47(3):391-400.
11. Stallman HM. Psychological distress in university students: A comparison with general population data. *Australian Psychologist*. 2010;45(4):249-57.
12. Robinson CM, Klenck SC, Norton PJ. Psychometric properties of the Generalized Anxiety Disorder Questionnaire for DSM-IV among four racial groups. *Cognitive behaviour therapy*. 2010;39(4):251-61.

Appendix C

Dear Editor,

We were delighted to receive such positive appraisals of our revision to the manuscript '**Pawsitively sad: Pet-owners are more sensitive to negative emotion in animal distress vocalisations**'. We thank the reviewers for the comments, 'Your article has been greatly improved by your answers to the reviewers, and it is now very pleasant to read it. I appreciate the additional online survey you did, and all the efforts to revise the text', 'The authors did a good job in reorganizing the manuscript and reaching a more balanced conclusion' and 'It was a pleasure to review this interesting study'.

We have addressed Reviewer 1's suggestion of adding data on the human and baby crying to the Supplementary materials (S.Table 1 and S. Figure 3). This reviewer also noted some inconsistencies in decimal places, which we have corrected throughout.

We look forward to seeing our work published in Royal Society Open Science.

Yours sincerely,

Christine Parsons (on behalf of the authors)

Christine Parsons
Associate professor

Interacting Minds Centre
(IMC)

Christine Parsons
Associate professor

Date: 3 July 2019

Direct Tel.: +45 8716 2127
E-mail:
christine.parsons@cas.au.dk
Web:
[http://pure.au.dk/portal/en/
christine.parsons@cas.au.dk](http://pure.au.dk/portal/en/christine.parsons@cas.au.dk)

Sender's CVR no.: 31119103

Page 1/1